# Defining the Immune Checkpoint Landscape in Human Colorectal Cancer Highlights the Relevance of the TIGIT/CD155 Axis for Optimizing Immunotherapy

**DOI:** 10.3390/cancers14174261

**Published:** 2022-08-31

**Authors:** Kathleen Ducoin, Linda Bilonda-Mutala, Cécile Deleine, Romain Oger, Emilie Duchalais, Nicolas Jouand, Céline Bossard, Anne Jarry, Nadine Gervois-Segain

**Affiliations:** 1Nantes Université, Univ Angers, INSERM, CNRS, Immunology and New Concepts in ImmunoTherapy, INCIT, UMR 1302/EMR6001, F-44000 Nantes, France; 2LabEx IGO, Université de Nantes, F-44000 Nantes, France; 3Nantes Université, Univ Angers, INSERM, CNRS, Centre de Recherche en Cancérologie et Immunologie Intégrée Nantes-Angers, CRCI2NA, UMR 1307/EMR6001, F-44000 Nantes, France; 4Institut Roche, F-92100 Boulogne-Billancourt, France; 5CHU Nantes, Department of Digestive Surgery and IMAD, F-44000 Nantes, France; 6Cytocell, BioCore, Nantes Université UMS 3556, Inserm US016, CNRS UAR 3556, CHU Nantes, SFR Santé François BONAMY, F-44000 Nantes, France; 7CHU Nantes, Pathology Department, F-44000 Nantes, France

**Keywords:** immune checkpoints, TIGIT, tumor-infiltrating lymphocyte, colorectal cancer, immunotherapy

## Abstract

**Simple Summary:**

One promising avenue for the treatment of colorectal cancer is the reinvigoration of pre-existing anti-tumor T-cell responses by overcoming inhibitions that arise from the T-cell expression of immune checkpoints. This study aims to perform an in-depth assessment of human colorectal cancer, the immune checkpoint landscape in tumor-infiltrating T cells and their respective ligands in tumor cells and myeloid cells; this is achieved using multiparametric flow cytometry on dissociated fresh tumors from 40 patients and in situ immunohistochemistry. Our results highlight the strong expression of TIGIT and its ligand, CD155, in tumors, suggesting the role of the TIGIT/CD155 axis in colorectal cancer. Moreover, unsupervised clustering allowed us to identify two distinct subgroups of colorectal cancer patients according to the co-expression of several immune checkpoints on T-lymphocyte subpopulations. Altogether, our findings support the development of colorectal cancer immunotherapies targeting TIGIT, which could be used in combination immune checkpoint therapy in colorectal cancer.

**Abstract:**

While immune checkpoint (IC) therapies, particularly those targeting the PD-1/PD-L1 axis, have revolutionized the treatment of melanoma and several other cancers, their effect remains very limited in colorectal cancer (CRC). To define a comprehensive landscape of ICs in the human CRC tumor microenvironment (TME), we evaluated, using multiparametric flow cytometry, their ex vivo expression via tumor-infiltrating lymphocytes (TILs) (n = 40 CRCs) as well as that of their respective ligands on tumor and myeloid cells (n = 29). Supervised flow cytometry analyses showed that (i) most CD3^+^ TILs expressed PD-1 and TIGIT and, to a lesser extent, Tim-3, Lag3 and NKG2A, and (ii) EpCAM^+^ tumor cells and CD11b^+^ myeloid cells differed in their IC ligand expression profile, with a strikingly high expression of CD155 by tumor cells. An in situ analysis of IC and their ligands using immunohistochemistry on paraffin sections of CRC confirmed the overexpression of TIGIT and its ligand, CD155, in the TME. Most interestingly, an unsupervised clustering analysis of IC co-expression on CD4^+^ and CD8^+^ TILs identified two tumor subgroups, named IC^high^ and IC^low^. Altogether, our findings highlight the TIGIT/CD155 axis as a potential target that could be used in combination IC therapy in CRC.

## 1. Introduction

Colorectal cancer (CRC) is the world’s second most fatal cancer, comprising approximately 9.4% of cancer-related deaths in 2020 [1]. Despite improved screening methods and recent advances in systemic and local treatments, CRC remains the leading cause of cancer-related death worldwide. While distant metastases (associated with stage IV) are the leading cause of CRC-related mortality, patients with stage II colon cancer are a heterogeneous population for whom there is no single answer as to whether or not chemotherapy should be administered, and the nature of chemotherapy is still controversial [2]. Therefore, there is a need to invest in the research and development of new therapies more appropriate for these patients.

American Joint Committee on Cancer/Union Internationale Contre le Cancer (AJCC/UICC) TNM staging, based on the histopathologic criteria of tumor invasion—namely the extent of the primary tumor (T) and the spread to the lymph nodes (N) or distant metastases (M)—provides the current guidelines for the classification of CRCs. However, the clinical outcome can be very different between patients at the same stage. More recently, a classification named Consensus Molecular Subtypes (CMS) has been defined based on the mutational status on specific genes, epigenetic signatures and differential gene expression in tumor tissues, including both cancer cells and the tumor microenvironment (TME). CRCs are, thus, divided into four subtypes (CMS1-4); however, 13% are unclassifiable CRCs due to mixed features [3]. CMS1 (14%) is an immunogenic subtype, mostly enriched in MSI tumors and *BRAF*-mutated tumors; CMS2 (37%) has epithelial features with marked WNT and MYC signaling and elevated chromosomal instability; CMS3 (13%) is enriched in *KRAS*-mutated tumors and has obvious metabolic dysregulation; and CMS4 (23%) is a mesenchymal subtype characterized by prominent TGF-β activation, stromal invasion, angiogenesis and an inflammatory and immunosuppressive phenotype. Although this classification has clear potential for clinical use in predicting both the prognosis and the response to systemic therapy, it is not routinely implemented in clinical practice [4].

One of the promising avenues in the treatment of CRC (and many other cancers) takes into account the TME, which is composed of diverse and heterogeneous immune infiltrations, whose densities are associated with prognostic significance [5]. Indeed, a new definition of cancer has emerged, involving, at all stages, complex and dynamic interaction between the tumor cells and the immune system. This has led to the definition of the immune contexture, which represents the pre-existing immune parameters and is associated with patient survival. The Immunoscore (IS), introduced [6] and defined by Jérome Galon and colleagues [7], is a scoring system (from I0 to I4) based on immunohistochemistry and digital pathology; it assesses the in situ densities of two lymphocyte populations, CD3^+^ and CD8^+^ T cells, in the tumor and its invasive margin. Overall, the IS has been shown to be superior to the AJCC/UICC TNM classification for CRC patients [8]. It is a significant consensus prognostic marker of survival in stage I–III CRC patients, as well as specifically for stage II and III patients. Furthermore, Williams et al. defined, in a large cohort of stage II/III CRC patients, a combined TIL (tumor-infiltrating lymphocyte)/MMR (mismatch repair)-based classification that distinguishes tumors prognostically [9]. Even so, the role of distinct immune cell types, especially CD8^+^ T cells, in modulating cancer progression is increasingly emerging.

However, during cancer progression, it has been shown that TILs exhibit an increase in the chronic expression of various immune checkpoints (ICs), resulting in the functional exhaustion and unresponsiveness of T cells. IC inhibitors (ICis) could boost the antitumor response. In this context, numerous clinical trials have been conducted in patients with microsatellite instability-high metastatic CRC (MSI-H mCRC), demonstrating the efficiency of anti-PD-1 (programmed cell death 1) mAbs; this includes pembrolizumab, which has been approved by the FDA as an effective first-line therapy for these patients, and which was superior to chemotherapy with respect to progression-free survival in patients with MSI-H mCRC [10]. The combination of ICis has also been applied in patients with MSI-H mCRC, and is responsive to the anti-PD-1 mAb nivolumab combined with low doses of ipilimumab, an anti-CTLA-4 (cytotoxic T-lymphocyte-associated protein 4) mAb [11]. Nonetheless, these therapies concern a limited number of CRC patients and there is still a need to identify additional IC targets for immunotherapy for the majority of CRCs.

It has been reported that IC expression negates the prognostic relevance of cytotoxic T-lymphocytes in highly immunogenic CRC tumors and predicts a poor outcome in patients with MSI CRC [12,13]. For example, the upregulation of Tim-3 (T-cell immunoglobulin and mucin domain 3), PD-1, TIGIT (T-cell immunoglobulin and immunoreceptor tyrosine-based inhibitory motif domain) or Lag3 (Lymphocyte activation gene 3) has been associated with poor prognosis in CRC patients [14,15,16,17,18]. While the expression of ICs in CRC has been addressed in several reports, these are generally based on transcriptomic studies that do not strictly reflect the protein expression of these ICs. A recent study reported, using flow cytometry analysis, the expression, in descending order, of TIGIT, PD-1 and Tim-3 by the TILs (CD4^+^ and CD8^+^) of CRC (and that of their ligands by tumor cells) [19]. Nevertheless, this study was performed on a relatively small cohort of 20 patients and requires confirmation and deepening. Moreover, these studies do not systematically include all ICs, particularly one of the most recently described members of this family, namely NKG2A (Natural Killer group protein 2A) [20], which we have previously shown to negatively impact the anti-tumor CD8 T-cell response in CRC [21].

Therefore, it is crucial to define the complete mapping of ICs expressed by CD3^+^ TILs in CRC in order to better stratify CRC patients and further personalize immunotherapy combining one or more ICis, and to avoid both the under- and over-treatment of CRC patients.

The main objective of this study was to simultaneously analyze, using flow cytometry, the relative ex vivo expression of five major ICs, namely PD-1, TIGIT, Tim-3, Lag3 and NKG2A, by the CD3^+^ TILs of CRCs, as well as that of their ligands by the tumor cells and myeloid cells of the TME. Reinforced by the immunohistochemical study of the in situ expression of ICs and their ligands in CRC sections, our results highlight the TIGIT/CD155 axis as the most likely to inhibit the majority of anti-tumor T responses. In addition, and most interestingly, unsupervised clustering in CD4^+^ and CD8^+^ TILs, according to their IC expression level, allowed us to identify IC^high^ and IC^low^ subgroups that should better define a patient’s eligibility for combined immunotherapy using several ICis.

## 2. Materials and Methods

### 2.1. Patients

Forty patients undergoing surgery for CRC without prior chemotherapy or radiotherapy, at the University Hospital of Nantes or the “Nouvelles Cliniques Nantaises” (France), were included in this study; they were divided into three “cohorts” according to the improvements made over time to the multiparametric flow cytometry panel used for the study of IC expression by TILs (see Section 2.5). The pathological staging of all patients was assessed according to the eighth edition of the TNM (Tumor Node Metastasis) staging system for CRC published by the International Union against Cancer (UICC eighth edition). The histological subtyping was reviewed according to the fourth edition of World Health Organization classification of tumors of the digestive system. All tissues were processed according to the Helsinki declaration and the guidelines of the French Ethics Committee for research on human tissues. Our tissue biocollection was registered with the French Ministry for Higher Education and Research (DC-2014-2206) with approval from the ethics committee (CPP Ouest IV, Nantes). Each patient included in this study signed an informed consent form. Table 1 lists the clinicopathological features of the 40 CRC patients derived from the three “cohorts”.

### 2.2. CMS Classification of CRC

The CMS classification of 31 CRC tumors was obtained using 3′ RNA sequencing, as previously described [21]. Briefly, tissue homogenization (using Fastprep-24 (MP Biomedicals, Irvine, CA, USA)), RNA extraction (RNeasy Mini Kit (Qiagen, Hilden, Germany)), the evaluation of total tumor RNA amounts (NanodropND-100 spectrophotometer (Thermo Fisher Scientific, Waltham, MA, USA) and quality evaluation (2100 Bioanalyzer instrument (Agilent Technologies, Santa Clara, CA, USA)) were performed, before using the QIAseq UPX 3′ transcriptome kit (Qiagen) to generate transcriptomic data. Primary analysis was then performed using GeneGlobe (Qiagen, www.qiagen.com/GeneGlobe, accessed on 2 September 2020), and the R package CMS classifier was used to determine the CMS classification of the CRC tumors.

### 2.3. Immunohistochemistry and Immunostaining Scores

Immunostaining was performed on 3 μm formalin-fixed, paraffin-embedded sections from a representative tumor block to assess (i) the CD3 and CD8 T-cell densities within the tumor and in peritumoral stroma, (ii) the expression of ICs by immune cells in both regions, (iii) the expression of some IC ligands by tumor cells and immune cells and (iv) the MSS (microsatellite stability)/MSI status of the tumor. The following primary antibodies were used: CD3 (Agilent Technologies, polyclonal rabbit anti-human, RRID:AB_2335677), CD8 (Agilent Technologies, monoclonal mouse anti-human clone C8/144B, RRID:AB_2075537), PD-1 (Abcam, Cambridge, UK, monoclonal mouse anti-human clone NAT105, RRID:AB_881954), TIGIT (Cell Signaling Technology, Danvers, MA, USA; monoclonal rabbit anti-human clone ESY1W, RRID:AB_2922806), Tim-3 (Bio-Techne, Minneaoplis, MN, USA, polyclonal goat anti-human, RRID:AB_355235), Lag3 (Novus, Bio-Techne, monoclonal mouse anti-human clone 17B4, RRID:AB_11162489), CD94 (Diaclone, Besançon, France; monoclonal mouse anti-human clone B-D49, RRID:AB_2922808), PD-L1 (Programmed death-ligand 1) (Cell Signaling Technology, monoclonal rabbit anti-human clone E1L3N, RRID:AB_2687655), CD155 (Cell Signaling Technology, monoclonal mouse anti-human clone D8A5G, RRID:AB_2799970), galectin-9 (Abcam, polyclonal rabbit anti-human, RRID:AB_1268942), HLA-DR (Abcam, monoclonal mouse anti-human clone SPM289, RRID:AB_444051) and HLA-E (Bio-Rad, Marnes-la-Coquette, France; monoclonal mouse anti-human clone MEM-E/02, RRID:AB_324025).

Immunostaining was performed using an automated stainer (AutostainerLink48, Agilent Technologies) according to a standard protocol, including antigen retrieval. The immunological reaction was visualized using a peroxidase/diaminobenzidine Envision system (Agilent Technologies), and the sections were counterstained with hematoxylin. The slides were then scanned using a NanoZoomer (Hamamatsu Photonics, Massy, France). The number of CD3^+^ and CD8^+^ TILs and of cells expressing the various ICs were counted per 100 tumor cells or stromal cells using QuPath (RRID:SCR_018257), an open-source software for digital pathology analysis. The counts were performed on 3 areas from 1 section of a tumor (0.3 mm^2^ to 1.5 mm^2^). Regions of interest were drawn (tumor glands and peritumoral stroma near the invasive margin). In each region (tumor and stroma), a total number of 5000 cells were counted in the 3 areas per section, and the results are expressed as the mean of the 3 counts. Concerning the IC ligands (PD-L1, CD155, galectin-9, HLA-DR and HLA-E), they were scored (from 0 to 3, Table 2) in the tumor cells and in the immune cells of the stroma according to a previously reported semi-quantitative score [22].

The MMR status of CRC was assessed via immunohistochemical detection of the MMR proteins on the paraffin sections using the following antibodies from Agilent Technologies: MLH1 (clone ES05, RRID:AB_2877720), MSH2 (clone FE11, RRID:AB_2889974), MSH6 (clone EP49, RRID:AB_2889975) and PMS2 (clone EP51, RRID:AB_2889977), as previously described [23]. The complete loss of expression of one or more MMR proteins indicates an MMR-deficient tumor corresponding to an MSI status.

### 2.4. Mechanical Dissociation of Colorectal Tissues

Fresh tumor samples were recovered using MACS Medium Tissue Storage Solution (Miltenyi Biotec, Bergisch Gladbach, Germany) and were mechanically dissociated before flow cytometry analyses, as previously described [23]. Briefly, the tissues were minced into small fragments at room temperature in RPMI 1640 medium and transferred to a GentleMACS C tube (Miltenyi Biotec) for non-enzymatic mechanical dissociation using a GentleMacs Dissociator (Miltenyi Biotec). After filtration through a 40 µm cell strainer (Dutscher, Bernolsheim, France), the suspension was centrifuged and resuspended in culture medium.

### 2.5. Ex vivo Flow Cytometry Staining

A total of 1 × 10^6^ cells from fresh dissociated tumor tissues were centrifuged, and then, incubated in tubes for each multiparametric panel in 100 µL of PBS with 2.5 µg of Human BD Fc Block (BD Biosciences, San-Jose, CA, USA; RRID:AB_2869554) to block the non-specific binding of fluorescent antibodies to cellular Fc-γ receptors. After 10 min at room temperature and centrifugation, the cells were incubated in 150 µL of Brilliant Stain Buffer (BD Biosciences, RRID:AB_2869750) containing appropriate concentrations of specific or isotype control Abs for 30 min at 4 °C.

Because of successive improvements in the multiparametric panel designed to determine the expression profile of ICs by CD3^+^ TILs (the substitution of anti-CD94 mAb with an NKG2A-specific mAb, or the use of mAbs coupled with optimized fluorochromes), the 40 CRC patients included in this study were divided into three cohorts. For the first cohort (n = 8), the following mAbs were used: CD3-APC-H7 (BD Biosciences, SK7, RRID:AB_1645730), CD4-BV786 (BD Biosciences, L200, RRID:AB_2738485), CD8-APC (BioLegend, San Diego, CA, USA; RPA-T8, RRID:AB_314132), PD-1-BV421 (BD Biosciences, EH12.1, RRID:AB_11153482), TIGIT-PerCPeFluor710 (eBioscience, Thermo Fischer Scientific, MBSA43, RRID:AB_10853679), Tim-3-BV650 (BD Biosciences, 7D3, RRID:AB_2722547), Lag3-PECy7 (eBioscience, 3DS223H, RRID:AB_2573430) and CD94-FITC (BD Biosciences, HP-3D9, RRID:AB_396200). For the second cohort (n = 20), the following mAbs were used: CD3-BUV395 (BD Biosciences, UCHT1, RRID:AB_2744387), CD4-BUV496 (BD Biosciences, SK3, RRID:AB_2744422), CD8-APC (BioLegend, RPA-T8, RRID:AB_314132), PD-1-BV421 (BD Biosciences, EH12.1, RRID:AB_2739399), TIGIT-PECy7 (BioLegend, A15153G, RRID:AB_2632929), Tim-3-PE (BioLegend, F38-2E2, RRID:AB_2116576), Lag3-BV650 (BioLegend, 11C3C65, RRID:AB_2632951) and NKG2A-AF488 (R&D systems, 131411, RRID:AB_2921333). For the third cohort (n = 12), only the mAb NKG2A-AF488 was replaced by NKG2A-FITC (Miltenyi Biotec, REA110, RRID:AB_2733623).

The second multiparametric panel, designed to investigate IC ligand expression in EpCAM^+^ tumor cells and CD11b^+^ myeloid cells, used the following anti-human mAbs for both cohorts 2 and 3: EpCAM-FITC (BioLegend, 9C4, RRID:AB_756077), CD11b-BV421 (BD Biosciences, ICRF44, RRID:AB_2737689), PD-L1-BUV395 (BD Biosciences, MIH1, RRID:AB_2740056), CD155-APC (BioLegend, SKII.4, RRID:AB_2565815), galectin-9-PE (BD Biosciences, 9M1-3, RRID:AB_2739386), HLA-DR-BV650 (BioLegend, L243, RRID:AB_2563828) and HLA-E-PECy7 (BioLegend, 3D12, RRID:AB_2565262).

Isotypic controls were also performed for each multiparametric panel with antibodies from the same manufacturer. For each panel, Fixable Viability Stain 780 (BD Biosciences, RRID:AB_2869673) was used to exclude dead cells, and staining was performed at the same time as the antibodies. After three washes in PBS 0.1% BSA, the stained cells were acquired in the viable cell gate on a BD LSRFortessa X-20 flow cytometer. The cytometry data were acquired using BD FACSDiva 8.0 software (BD Biosciences, RRID:SCR_001456). To standardize flow cytometry data acquisition from one CRC tumor to another, Rainbow Calibration Particles (8 peaks) (BD Biosciences, RRID:AB_2869258) were used.

### 2.6. Supervised and Unsupervised Analyses of Flow Cytometry Data

To analyze flow cytometry data, both FlowJo software 10.8.1 (BD Biosciences, RRID:SCR_008520) (supervised analysis) and the OMIQ online platform (https://www.omiq.ai/, accessed on 19 May 2022) (unsupervised analysis) were used. Prior to analysis, either supervised or unsupervised, the flow cytometry data were first cleaned using the FlowAI algorithm [24] with default settings. After confirmation of the data cleaning (or the modification of some settings to exclude bad events), the compensation was checked and applied on the samples. The gating strategy for the supervised analysis is shown in Appendix A, as well as for the unsupervised analysis, to focus on the population of interest.

For the unsupervised analysis, live CD3^+^, CD4^+^ and CD8^−^ singlet events and CD3^+^, CD8^+^ and CD4^−^ singlet events were pre-gated for each CRC tumor sample; then, they were subsampled to analyze the same number of events in all CRC samples of each cohort. The FlowSOM algorithm was run on the OMIQ platform for both cohort 2 and cohort 3, with the recommended optimal settings [25], i.e., Euclidean distance to find the nearest neighbor of a new point, a 10 × 10 grid for the self-organizing map (SOM), and the number of training iterations set to 10. Using the OMIQ platform, no prior concatenation of samples from each cohort was required. For the CD4^+^ cell unsupervised analysis, the PD-1-BV421, TIGIT-PECy7 and Tim-3-PE parameters were used to run the FlowSOM algorithms, and for the CD8^+^ cells unsupervised analysis, the PD-1-BV421, TIGIT-PECy7, Tim-3-PE, Lag3-BV650 and NKG2A-AF488(-FITC) parameters were used. To visualize the distribution of FlowSOM-generated clusters, all UMAP dimension reductions were also run on the OMIQ platform using the default settings: 0.4 (or 0.9 for the UMAPs of the CD8^+^ population of cohorts 2 and 3) as the minimum distance, and 15 as the nearest neighbors and Euclidean distance, with the same parameters as listed for the FlowSOM algorithms. The “Clustered heatmap” tool on the OMIQ platform was used to generate heatmaps of the medians of the ICs expressed by CD4^+^ cell clusters and CD8^+^ cell clusters.

Principal component analyses (PCA) were generated using RStudio Desktop (version 2022.07.0 + 548, RRID:SCR_000432) by using the factoextra package.

### 2.7. Statistical Analyses

The statistical analyses were performed using GraphPad Prism 8 software (RRID:SCR_002798). The Mann–Whitney test, Wilcoxon paired *t*-test and Kruskal–Wallis test, followed by Dunn’s multiple comparisons test, were performed in this study and are specified in each figure legend. A *p*-value < 0.05 was considered statistically significant (*), and *p* < 0.01 (**), *p* < 0.001 (***) and *p* < 0.0001 (****) as highly significant.

## 3. Results

### 3.1. IC Expression by T Cells Infiltrating CRC Tumors

To estimate the potential impact of ICs on the antitumor T-cell responses in CRC, we analyzed, using flow cytometry, the ex vivo expression of five of them—namely PD-1, TIGIT, Tim-3, Lag3 and NKG2A—in CD3^+^ TILs, present in 40 dissociated CRC tumors (three cohorts, see Table 1), including 34 MSS and 6 MSI tumors, with the gating strategy shown in Appendix A.

Since no differences were observed between the frequencies of the CD3^+^ TILs expressing each of the ICs among the three cohorts, particularly between the CD94^+^ CD3^+^ TILs in cohort 1 and the NKG2A^+^ CD3^+^ TILs in cohorts 2 and 3, we pooled all the data. As shown in Figure 1A,B, a high proportion of CD3^+^ TILs expressed PD-1 (median values: 77% in MSS vs. 92.4% in MSI) and TIGIT (83.3% in MSS vs. 90.1% in MSI), and a lower proportion expressed Tim-3 (28.8% in MSS vs. 66.9% in MSI), Lag3 (13.3% in MSS vs. 21.6% in MSI) and CD94-NKG2A (4.7% in MSS vs. 15.3% in MSI). Interestingly, the median frequency of all these ICs on the CD3^+^ TILs was higher in MSI compared to MSS CRCs, with statistical significance for PD-1, TIGIT and Tim-3 (*p* = 0.0017, *p* = 0.0131 and *p* = 0.0059, respectively). In addition, our results highlighted a subgroup of MSS CRCs with higher expression of ICs (close to that of MSI tumors) and a few MSI CRCs with weaker expression of ICs (Figure 1B). These observations were confirmed when considering the relative expression of each IC in a given CRC patient (Figure 1C).

When examining the median fluorescence intensity (MFI) of IC-positive cells among the CD3^+^ TILs (data from cohorts 2 and 3 that have equivalent levels of MFI, n = 32; 27 MSS, 5 MSI), the results obtained mirror those of frequencies, with higher expression levels of PD-1, TIGIT, Tim-3, Lag3 and NKG2A in descending order (Figure 1D). Furthermore, these MFIs were higher in MSI than in MSS tumors with statistically significant differences for PD-1 (2805 in MSI vs. 1728 in MSS, *p* = 0.0221) and TIGIT (2043 in MSI vs. 1265 in MSS, *p* = 0.0258). Interestingly, patients with a high frequency of CD3^+^ TILs expressing PD-1, TIGIT and Tim-3 also had the highest MFI of these ICs (data not shown).

Next, we examined the IC expression level, both in terms of frequency (n = 40, Appendix A) and MFI (n = 32, Appendix A), in relation to the pTNM stage in CRC patients from the three cohorts (stage I to III, no stage IV). We observed an overall trend towards an increase in all five ICs in stage II CRC and a decrease in stage III, although these trends were not statistically significant (except for Tim-3 in terms of frequency, *p* = 0.011), probably due to the high inter-patient variability within the pTNM stages.

We also examined IC expression levels in relation to the CMS classification of CRC, which could be determined for 31 tumors in the three cohorts using the R CMS classifier package [3]. The percentages of CRCs within each subgroup were similar to those described in the literature; however, there was enrichment of the CMS1 tumors at the expense of CMS4 tumors, with seven CMS1 (22.6%), twelve CMS2 (38.7%), six CMS3 (19.3%), three CMS4 (9.7%) and three unclassified (9.7%) tumors (see Table 1). Due to heterogeneity within the CMS subtypes, it was difficult to find an obvious relationship between IC frequency or expression level and CMS classification. However, there was a statistical trend toward higher frequency and MFI in CD3^+^ TILs expressing ICs in CMS1 (containing mainly MSI tumors) versus the other subgroups (significant for Tim-3 (Dunn’s multiple comparisons test, *p* = 0.0382) and TIGIT (Kruskal–Wallis test, *p* = 0.0446)) (Appendix A).

Finally, we performed principal component analysis (PCA) of the expression level of the different ICs among the CD3^+^ TILs (from cohorts 2 and 3, n = 32), taking into account both the percentage of positive cells and the MFI for each of the five ICs in the MSS and MSI CRCs (Figure 1E). The results confirmed the existence of a subgroup of MSS CRCs close to most MSI CRCs, and conversely, of some MSI CRCs close to classical MSS CRCs.

Furthermore, we assessed the differential expression of the five ICs on the CD4^+^ versus the CD8^+^ TIL subpopulations. We first determined the proportion of CD4^+^ (CD8^-^) and CD8^+^ (CD4^−^) T cells among the CD3^+^ TILs in the tumors of the three cohorts combined. No statistical difference was found according to the different classifications of MSS/MSI (n = 40, Figure 1F), the pTNM stages (n = 40, Appendix A) and CMS (n = 28, Appendix A). Nevertheless, a slightly higher frequency of CD8^+^ T cells among the CD3^+^ TILs in MSI tumors was observed compared to MSS tumors (37.6% vs. 32%, *p* = 0.2427) and in stage II vs. stage III tumors (35.5% vs. 27.0%, *p* = 0.2319).

### 3.2. Identification of CRC Subgroups Based on IC Expression by CD4^+^ TILs

We then performed an unsupervised analysis of flow cytometry data, taking into account the expression levels (frequency and MFI) of PD-1, TIGIT and Tim-3 by CD4^+^ TILs. It should be noted that since the CD4^+^ TILs expressed neither Lag3 nor NKG2A (data not shown), these markers were excluded from this analysis. In cohort 2 (n = 20), we identified eleven CD3^+^ CD4^+^ TIL clusters whose characteristics are shown in Figure 2A–C. The most represented cluster (cluster 10, with a median frequency of 18.7% of total cells) corresponded to CD4^+^ TILs expressing intermediate (int) or high levels of the three ICs (PD-1^int^ TIGIT^high^ Tim-3^high^). Only a small proportion of CD4^+^ TILs (cluster 3, with a median frequency of 6.1% of total cells) did not express any of the ICs studied.

As the eleven identified clusters were confirmed in cohort 3 (n = 12) in quite similar proportions (Appendix A–C), we performed an overall analysis, on the two cohorts combined (n = 32), of the distribution of CD4^+^ TIL clusters according to the microsatellite status, pTNM stage and CMS classification of the CRCs.

As shown in Figure 2D, MSI CRC showed a significant increase in the percentage of cells belonging to the clusters displaying the highest levels of ICs compared to MSS CRC, i.e., cluster 10 (PD-1^int^, TIGIT^high^ and Tim-3^int^, median frequencies: 21.4% in MSI vs. 12.6% in MSS, *p* = 0.0455) and cluster 11 (PD-1^high^, TIGIT^high^ and Tim-3^high^, median frequencies: 12.8% in MSI vs. 2.2% in MSS, *p* = 0.0216). Conversely, a decrease in the percentage of cells belonging to the clusters with the lowest levels of ICs, i.e., cluster 2 (TIGIT^low^) and cluster 3 (no IC), was observed in MSI vs. MSS CRCs.

Considering pTNM staging, no significant differences were observed in the proportion of positive cells among clusters in the various pTNM stages (Appendix A). However, when examining the CMS subgroups, significant differences were observed in the frequency of CD4^+^ TILs in “IC^high^” clusters 10 (Kruskal–Wallis test, *p* = 0.0342) and 11 (Kruskal–Wallis test, *p* = 0.0464) in immunogenic CMS1 tumors encompassing the majority of MSI CRC (Appendix A).

We then constructed trajectories based on the increase in the expression of the three ICs by the different CD4^+^ TIL subgroups. We found two distinct curves starting from CD4^+^ T cells not expressing ICs (cluster 3) and converging toward cells strongly expressing the different ICs (clusters 10 and 11); one passed through the sequential expression of TIGIT, PD-1 then Tim-3 (clusters 2 then 1...), and the other through the sequential expression of PD-1, Tim-3 then TIGIT (clusters 4, 6 then 7…) (cohort 2 (n = 20) and cohort 3 (n = 12), Appendix A, respectively).

By deconstructing these results on a patient-by-patient basis and considering the microsatellite status, we clearly identified, in the two cohorts, five out of twenty-seven MSS tumors (18.5%) (C152, C162, C168, C170 and C173) with an IC expression profile close to that of MSI; conversely, we identified one out of five MSI tumors (20%) (C158) with a profile similar to that of MSS tumors (cohort 2 (16 MSS, 4 MSI), Appendix A and cohort 3 (11 MSS, 1 MSI), Appendix A, with outlier patients indicated by black arrows).

To confirm these outliers in MSS and MSI tumors, PCA was then performed according to the CD4^+^ TIL clusters identified on the basis of their IC signatures in the two cohorts (n = 32) (Figure 2E). Using the 2D distribution of all patients on the PCA, we drew two circles, each identifying a tumor group. The first circle (in pale green), corresponding to tumors displaying low IC expression in TILs, contained the majority of MSS tumors and one (out of five) MSI tumor (in total, 20/32, i.e., 62.5%); the second circle (in blue), corresponding to tumors displaying the highest expression of ICs, contained the majority of MSI tumors as well as eight (out of twenty-seven) MSS tumors (a total of 12/32, i.e., 37.5%). These “MSS-like” or “MSI-like” tumors, named “IC^low^” or “IC^high^” tumors, respectively, were similar to those identified using UMAP analysis; however, there were two additional MSS tumors in the MSI-like group identified using PCA (C151 and C197).

### 3.3. Identification of CRC Subgroups Based on IC Expression by CD8^+^ TILs

Using the same approach and taking into account the expression of the five ICs by the CD8^+^ TILs, fourteen CD8^+^ TIL clusters were identified in cohort 2 (n = 20) (Figure 3A–C). The two most represented clusters were cluster 1 (with a median frequency of 12.8% of the total cells), corresponding to CD8^+^ TILs expressing a high level of PD-1, TIGIT, Tim-3 and Lag3, and cluster 12 (with a median frequency of 12.8% of the total cells), corresponding to CD8^+^ TILs expressing only intermediate levels of TIGIT and Lag3. It is noticeable that, in contrast to the CD4^+^ TILs, all CD8^+^ TILs express at least one IC, with cluster 14 expressing the fewest ICs (only low Lag3 expression, with a median frequency of 2.9% of the total cells).

As the fourteen identified clusters were confirmed in cohort 3 (n = 12) in quite similar proportions (Appendix A–F), global analyses of the distribution of CD8^+^ TIL clusters according to the microsatellite status, pTNM stage or CMS classification were performed on the two cohorts combined (n = 32).

The overall analysis of CD8^+^ TIL distribution in these clusters (n = 32) according to the microsatellite status (Figure 3D) revealed a trend towards an increase in MSI tumors in the percentage of cells belonging to the two clusters characterized by stronger IC expressions (cluster 1, PD-1^high^, TIGIT^high^, Tim-3^high^ and Lag3^high^, and cluster 6, PD-1^high^, TIGIT^high^, Tim-3^high^, Lag3^high^ and NKG2A^int^).

No differences were observed in the proportion of the different clusters according to the pTNM stage of CRC (Appendix A) or according to their CMS classification, although there was a tendency for the frequency of CD8^+^ TILs to increase in CMS1 immunogenic tumors in clusters 1 and 6, characterized by high IC expression (Appendix A).

We then projected the CD8^+^ TIL clusters onto the generated UMAP and observed different paths, starting from clusters expressing few ICs (cluster 13, PD-1^int^ and Lag3^int^, and cluster 14, Lag3^low^) and converging to clusters strongly expressing the different ICs (cluster 1, PD-1^high^, TIGIT^high^, Tim-3^high^ and Lag3^high^, and cluster 6, PD-1^high^, TIGIT^high^, Tim-3^high^, Lag3^high^ and NKG2A^int^) (cohort 2 (n = 20) and cohort 3 (n = 12), Appendix A, respectively).

By deconstructing these results on a patient-by-patient basis and considering the MSS/MSI status, we distinctly identified four tumors (C161, C162, C173 and C189) out of twenty-seven MSS tumors (14.80%) with an IC expression profile closer to that of MSI; conversely, we identified one (C158) out of five MSI tumors (20%) with a profile similar to that of MSS tumors (cohort 2 (sixteen MSS, four MSI), Appendix A, and cohort 3 (eleven MSS, one MSI), Appendix A, with patient outliers indicated by black arrows).

To confirm these outliers in MSS and MSI tumors, PCA was then performed according to the CD8^+^ TIL clusters identified on the basis of their IC signature in the two cohorts (n = 32) (Figure 3E). Similar to the analysis of CD4^+^ TILs, we drew two circles, each defining a group of tumors. The first circle (pale green) contained the majority of MSS tumors and two (out of five) MSI tumors (in total, 25/32, i.e., 78.1%), and the second (dark blue) contained the majority of the MSI tumors as well as four (out of twenty-seven) MSS tumors (in total, 7/32, i.e., 21.9%). These IC^low^ and IC^high^ tumors are the same as those identified using the UMAP analysis, with one more MSI tumor in the IC^low^ group (C194).

### 3.4. In Situ T-Cell Density and IC Expression by TILs in the Subgroups of CRC Identified Using Flow Cytometry

We then performed a quantitative in situ assessment of the CD3^+^ and CD8^+^ T cells, as well as IC-expressing cells, using immunohistochemistry on paraffin sections from the tumors of cohorts 2 and 3 (n = 31, 26 MSS, 5 MSI), both inside the tumor glands and in the peritumoral stroma. Both the T-cell infiltrate (CD3^+^ and CD8^+^) and the IC-expressing cells were higher in the peritumoral compared to the tumor compartments. We then compared the IC^low^ (n = 24) to the IC^high^ (n = 7) subgroups identified using flow cytometry, based on IC expression by the CD8^+^ TILs. The density of the CD3^+^ and CD8^+^ TILs, which was heterogeneous among patients, was slightly higher in the IC^high^ than the IC^low^ subgroup, both inside the tumor and in the peritumoral stroma (Figure 4B, medians: 1.3% vs. 0.8% CD3^+^ TILs and 0.8% vs. 0.5% CD8^+^ TILs, and 7.6% vs. 6.4% CD3^+^ cells and 2.9% vs. 2.2% CD8^+^ cells in the stroma in the IC^high^ vs. IC^low^ subgroups, respectively). Representative images of CD3 and CD8 immunostaining of the IC^high^ subgroup (an MSS CRC) are shown in Figure 4A. Regarding IC expression, positive cells could be quantified for PD-1, TIGIT and CD94; however, they could not be quantified for Lag3 as it was barely expressed by TILs in almost all cases, or Tim-3, which was difficult to quantify—specifically for lymphocytes—as it was also expressed by other immune cells. As shown in Figure 4C,D (representative images of an IC^high^ CRC and box plots), TIGIT^+^ and CD94^+^ cell numbers were significantly higher inside the tumor in the IC^high^ than in the IC^low^ subgroup (medians: 1.0% vs. 0.1% TIGIT^+^ cells and 0.8% vs. 0.5% CD94^+^ cells, and 1.1% vs. 0.3%; *p* values 0.005 and 0.013, respectively), whereas PD-1^+^ cell numbers were rather low in the two subgroups, except in some MSI and two MSS CRCs. In the stroma, the results follow the same trend.

### 3.5. Expression of IC Ligands by Tumor Cells and Myeloid Cells in CRC

As the presence of ICs cannot impact T-cell immune responses without the presence of their respective ligands, when sufficient tumor material was available, we assessed, using flow cytometry, the frequency of the ex vivo expression of PD-L1, CD155, galectin-9, HLA-DR and HLA-E (the ligands of PD-1, TIGIT, Tim-3, Lag3, and NKG2A, respectively), not only by EpCAM^+^ tumor cells but also by CD11b^+^ myeloid cells present within the TME. The gating strategy is shown in Appendix A. First, the ex vivo analysis of CRC tumors from cohorts 2 and 3 (n = 29, 24 MSS and 5 MSI) showed a median frequency of 22.7% tumor cells (ranging from 6.9% to 79.7%) and a lower percentage of myeloid cells with a median frequency of 3.4% (ranging from 0.1% to 26.7%) among viable cells (Figure 5A).

Interestingly, a different expression profile of IC ligands was observed between tumor cells and myeloid cells (Figure 5B, representative histograms and Figure 5C). Strikingly, the frequency of tumor cells expressing CD155 was much higher than that of myeloid cells (median frequency of 65.2% vs. 6.7%, *p* < 0.0001). By contrast, PD-L1 expression was low both in tumor cells and myeloid cells (2.5% and 1.8%, respectively). Although there is great heterogeneity in the frequency of EpCAM^+^ cells expressing HLA-DR and HLA-E (ranging from 2.1% to 95.9% and 0.5% to 95.6%, respectively), a higher proportion of tumor cells expressed these ligands compared to myeloid cells (median frequency of 29.5% vs. 16.1%, *p* = 0.0326 and 16.3% vs. 5.2%, *p* = 0.0009, respectively). Galectin-9 was expressed by a small proportion of tumor and myeloid cells, with a trend toward higher expression in tumor cells due to the existence of two tumors overexpressing this molecule (5.1% vs. 2.3%, *p* = 0.0534).

The in situ assessment of these ligands using immunohistochemistry (n = 31) confirmed the lack of PD-L1 expression in tumor cells in the vast majority of CRCs and, by contrast, the very strong expression of CD155, with strong membrane staining of tumor cells in about 80% of CRCs with a high score (score 3: more than 50% positive cells) (Figure 5D,E). PD-L1 and galectin-9 (membrane and cytoplasmic staining) were highly expressed by tumor cells in more than 75% of CRCs (score 3). HLA-E (membrane staining) was expressed in tumor cells more heterogeneously (Figure 5D,E). HLA-DR was expressed in only 30% of CRCs by tumor cells, with heterogeneous staining (mainly scores 1 and 2, i.e., 5 to 50% positive tumor cells). By contrast, HLA-DR was strongly expressed by myeloid cells within the stroma, mainly macrophages, surrounding the tumor glands (Figure 5D, representative image).

## 4. Discussion

The upregulation of inhibitory ICs by TILs or of their ligands by TME cells is one of the major mechanisms by which tumors escape host immunosurveillance. The objective of this study was to examine the landscape of ICs and their ligands in human CRC. We first explored, using flow cytometry, the ex vivo expression of five key inhibitory ICs by CD3^+^ TILs from CRC-dissociated tumors (n = 40) and showed that the vast majority (more than 80%) of these cells expressed not only PD-1 but also TIGIT (also reflected by a high MFI) and, at a lower frequency, Tim-3, Lag3, and then, NKG2A. These results are broadly in agreement with those in the literature reporting the expression of one or more of these TILs in CRC at the protein level [15,19,26,27] and using single cell RNA sequencing in CD4 and CD8 subpopulations [28]. Notably, very high frequencies of TILs expressing PD-1 and TIGIT have also been described in other cancers such as melanoma or head and neck squamous cell carcinoma [29,30]. In addition, our findings highlight a significantly higher frequency of TILs expressing not only PD-1 but also TIGIT and Tim-3; this is associated with an increased expression level in terms of MFI for TIGIT and PD-1 in MSI versus MSS CRCs, and accordingly, a slightly higher expression level in the CMS1 CRC encompassing mostly MSI CRC. These results strongly suggest that MSI tumors could benefit from ICi targeting PD-1, as well as TIGIT (and to a lesser extent Tim-3). We also observed a trend toward a higher expression level of the five ICs in stage II cancers, particularly compared to stage III; this finding is different from previous reports showing an increased frequency of TILs expressing these ICs in advanced stages of CRC (for PD-1 [15]; for TIGIT [16]; and for Tim-3 [14,26,31]). This discrepancy could be explained by the absence of stage IV tumors in our cohorts, which does not allow us to compare early stages (I/II) with advanced stages (III/IV) as in most studies. Furthermore, the overexpression of ICs by TILs is always associated with a poor prognosis in CRC patients (for PD-1 [15,16]; for TIGIT [16,17,27]; for Tim-3 [14,15,31]; and for NKG2A [23]).

To go one step further, we reanalyzed the flow cytometry data by separately examining the CD4^+^ and CD8^+^ TILs whose mean frequencies did not vary significantly according to MSS/MSI, staging or CMS classifications. Our results, showing a complete absence of Lag3 and NKG2A in the CD4^+^ subpopulation in contrast with the CD8^+^ subpopulation, are consistent with the literature [19,32]. The other three ICs were expressed by both CD4^+^ and CD8^+^ TILs, in line with other reports. Most interestingly, an unsupervised analysis of the flow cytometry data of the IC expression levels of CD4^+^ and CD8^+^ TILs led us to delineate a new classification of CRC. Indeed, based on the differential expression of these ICs, we identified 11 clusters for CD4^+^ TILs and 14 for CD8^+^ TILs, which led, via PCA analysis, to two subgroups of patients for each subpopulation. The first one, corresponding to tumors whose TILs show low expression of ICs (named the IC^low^ subgroup; 62.5% and 78.1% in the CD4^+^ and CD8^+^ subpopulations, respectively), contained the majority of MSS tumors and some MSI tumors. The second subgroup, corresponding to tumors whose TILs show the highest expression of ICs (named the IC^high^ subgroup; 37.5% and 21.9% in the CD4^+^ and CD8^+^ subpopulations, respectively), contained the majority of MSI tumors as well as some MSS tumors. These original findings extend previous data showing that some MSS tumors display features identical to those of MSI tumors, not only in terms of T-cell infiltrate/immune response and genomics but also in terms of IC co-expression levels [12,33,34]. In this sense, many reports have described the existence of MSS tumors with immune responses similar to that of MSI, especially in *POLE* (DNA polymerase epsilon, catalytic subunit)-mutated tumors, which are often associated with a high tumor mutation burden (TMB) and a strong immune infiltrate and, therefore, are likely to respond to ICis [33,35,36].

The in situ analysis of CRC tumor sections showed the presence of an intraepithelial CD8 T-lymphocyte infiltrate in both tumor subgroups, which was slightly more important within the IC^high^ tumors. Concerning IC expression, our in situ analysis paralleled the cytometry data and reinforced the presence of TIGIT-positive cells inside the tumor glands, and in contact with tumor cells, as well as in the peritumoral stroma near the tumor margin. The fact that these TIGIT^+^ TILs were significantly higher in the IC^high^ than the IC^low^ subgroup in contrast to PD-1^+^ TILs, whose density was identical in both subgroups, supports the relevance of targeting TIGIT and PD-1 in CRC tumors displaying a TIL infiltrate rich in both of these ICs. Furthermore, our results indicate that CD94^+^ intraepithelial TILs were significantly more numerous in the IC^high^ subgroup compared to IC^low^, reinforcing our previous findings [21,23] and the relevance of also targeting this checkpoint in combination therapy.

Interestingly, of the eight IC^high^ MSS tumors identified using PCA of CD4^+^ TILs, two were in common with the five identified for CD8^+^ TILs. Similarly, the only IC^low^ MSI tumor identified using PCA of CD4^+^ TILs was identical to one of the two identified for CD8^+^ TILs. These observations suggest specific mechanisms of the regulation of IC expression by CD4^+^ and CD8^+^ TILs. However, a few studies have provided some insights about IC regulation using transcriptional and epigenetic mechanisms; for example, the group of Elkord and colleagues, who showed that changes in the DNA methylation pattern and the enrichment of methylated histone marks in the promoter regions of IC genes might be important drivers of IC upregulation in the TME [37]. Specifically, they demonstrated that DNA hypomethylation and the repressive histones H3K9me3 and H3K27me3 are both involved in the upregulation of the *CTLA4* and *TIGIT* genes, whereas only the repressive histones are involved in the upregulation of the *PDCD1* and *HAVCR2* genes (coding for PD-1 and Tim-3) in CRC tumor tissues.

We then extended our study of the IC landscape by assessing, in parallel, their respective ligands in CRC, not only on tumor cells but also on myeloid cells, an issue that is less studied and reported. Our flow cytometry and immunohistochemistry data highlight differences in IC ligand expression between tumor cells and stromal immune cells. Among these, the most striking is the high expression of CD155 (TIGIT ligand) by tumor cells and a far lower expression by stromal immune cells. Conversely, PD-L1 was expressed by stromal immune cells and nearly absent from tumor cells, except in some MSI tumors with only a focal expression and in rare MSS CRCs. This result concerning PD-L1 is in line with previous reports [22,38] but not with others depicting high PD-L1 immunostaining in the tumor cells of CRC and, surprisingly, very weak expression using flow cytometry [19], reinforcing the importance of the standardization of immunohistochemistry among investigators (the PD-L1 antibody used, large tissue sections, a scoring system, etc.). Our results also showed overexpression of HLA-E (score 3) in about 20% of cases, which is in agreement with our previous data [39].

To date, most clinical trials in CRC have been conducted in patients with MSI-H mCRC and mainly targeted the PD-1/PD-L1 axis, demonstrating the efficacy of pembrolizumab [10] and nivolumab plus low-dose ipilimumab [11]. The KEYNOTE-177 investigators also showed that, in a randomized phase 3 trial, pembrolizumab resulted in significantly longer progression-free survival than chemotherapy when given as first-line therapy for MSI-H/dMMR mCRC, with fewer treatment-related adverse events [10]. However, since the percentage of CRC patients with MSI/dMMR (deficient MMR) is low, at around 12 to 15%, a significant proportion of CRC patients do not currently benefit from these ICis. A very recent study showed that stage II and III dMMR rectal tumors were very sensitive to dostarlimab, another anti-PD-1 mAb, with a complete clinical response for all patients included (n = 12) [40]. Following the accomplishments of PD-1 and CTLA-4 inhibitors, others targeting TIGIT, Tim-3, Lag3 and NKG2A are currently being explored in some solid tumors in various pre-clinical and clinical trials, in order to promote effective anti-tumor immunity with clinical benefits. The few results from phase I and II clinical trials targeting the TIGIT/CD155 axis, mainly in advanced non-small-cell lung cancer (NSCLC) patients [41,42], have demonstrated the efficacy of combining anti-TIGIT mAbs (vibostolimab or tiragolumab) with anti-PD-1 (pembrolizumab) or anti-PD-L1 (atezolizumab) mAbs, respectively. An ongoing trial is enrolling patients with locally advanced rectal tumors to evaluate the efficacy of tiragolumab plus atezolizumab after chemoradiotherapy (NCT05009069). In light of our results, we believe that the use of anti-TIGIT mAbs should be developed for colon- and rectal-cancer patients, not in combination with anti-PD-1 or -PD-L1 antibodies, but rather, in combination with other ICis according to IC expression levels. Among these, NKG2A could be a relevant candidate target due to its overexpression in about 15% of CD8^+^ TILs in CRC (often associated with HLA-E) and its strong impact on inhibiting anti-tumor T responses [21,23].

Our results highlight that (i) all MSI CRCs are likely to not be eligible for ICi treatments and, conversely, (ii) a large fraction of MSS CRCs are likely to respond to these immunotherapies. In this regard, recent preclinical and clinical studies have begun to show encouraging results, suggesting that treatment with ICis could be extended to an increasing number of MSS CRC tumors [43]. Although based on a limited number of patients, our findings strongly suggest that the TIGIT/CD155 axis can be targeted in therapies using ICi in CRC patients. In addition, the combination with other blockers of ICs, such as Tim-3 or NKG2A mAbs, should only be considered after a personalized analysis of IC expression in CRC patients. All these data, when validated in larger cohorts of CRC patients, will be useful for the development of promising strategies combining the administration of ICis with chemotherapy, molecular targeted therapy, radiotherapy or other new immunomodulatory agents [43].

## 5. Conclusions

In conclusion, using both ex vivo multiparametric flow cytometry combined with unsupervised clustering and in situ immunohistochemistry, this report provides an in-depth description of the landscape of ICs and their ligands in human CRC. Taken together, our findings highlight the TIGIT/CD155 axis in CRC and suggest that TIGIT should be considered as a relevant new target that could be used in combination IC therapy.

## Figures and Tables

**Figure 1 cancers-14-04261-f001:**
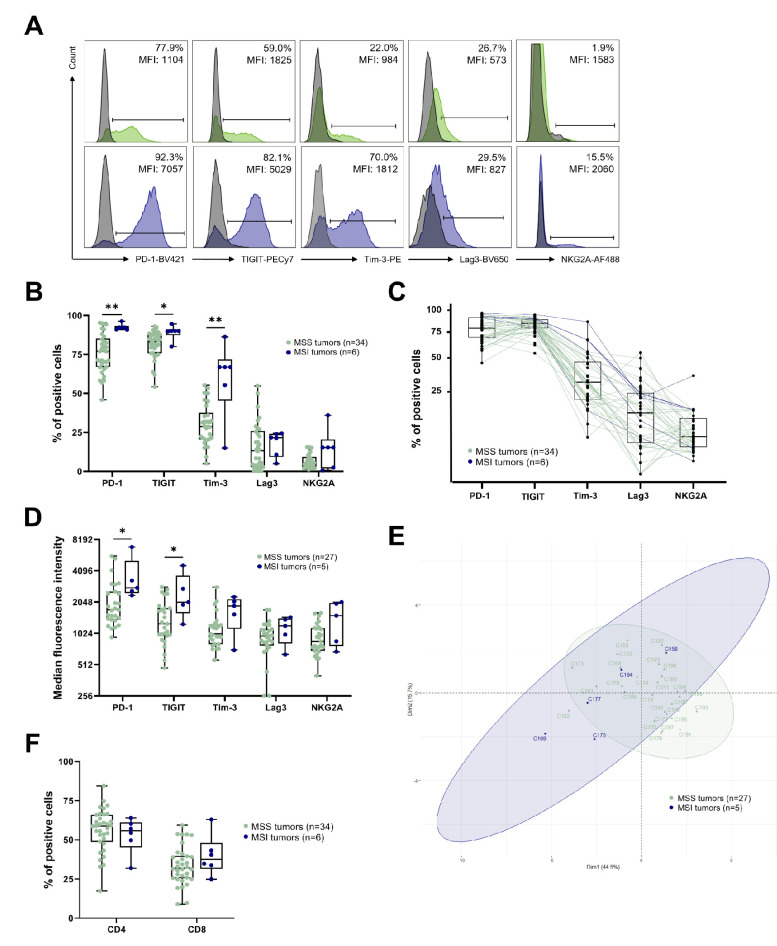
Expression of IC by CD3^+^ TILs from CRC tumors according to the microsatellite status. For each figure, data were analyzed according to the MSS/MSI status, with MSS tumors shown in pale green and MSI tumors in dark blue. (**A**) Representative histograms showing the expression of PD-1, TIGIT, Tim-3, Lag3 and NKG2A on CD3^+^ TILs from an MSS patient (upper panels) or from an MSI patient (lower panels). Results were expressed as positive cells and median fluorescence intensity (MFI). Isotypic controls are overlaid in grey. (**B**) Frequency of PD-1^+^, TIGIT^+^, Tim-3^+^, Lag3^+^ and NKG2A^+^ cells among CD3^+^ TILs (n = 40, cohorts 1, 2 and 3); Mann–Whitney test (* *p* < 0.05; ** *p* < 0.01). (**C**) Pairwise frequency of PD-1^+^, TIGIT^+^, Tim-3^+^, Lag3^+^ and NKG2A^+^ cells among CD3^+^ TILs (n = 40, cohorts 1, 2 and 3). (**D**) Expression level (MFI, scale in log2) of PD-1, TIGIT, Tim-3, Lag3 and NKG2A on positive IC CD3^+^ TILs (n = 32, cohorts 2 and 3); Mann–Whitney test (* *p* < 0.05). (**E**) Principal component analysis of tumors from cohorts 2 and 3 (n = 32) based on the frequency and MFI of the 5 ICs on CD3^+^ TILs. (**F**) Proportion of CD4^+^ and CD8^+^ cells among CD3^+^ TILs (n = 40, cohorts 1, 2 and 3); Mann–Whitney test.

**Figure 2 cancers-14-04261-f002:**
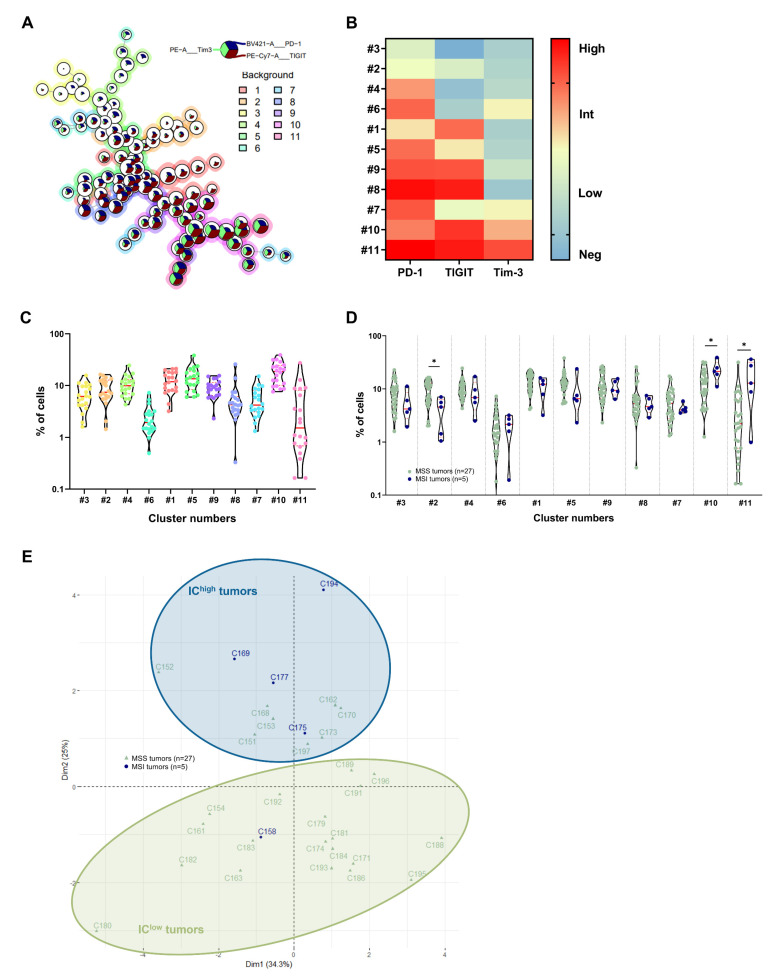
Identification of 11 CD4^+^ TIL clusters based on the differential expression of ICs. (**A**) FlowSOM tree of CD4^+^ TILs from cohort 2 (n = 20). The background color represents meta-clustering, and the legends of the star plot and meta-clustering are shown on the right side. (**B**) Heatmap of the MFI of the PD-1, TIGIT and Tim-3 markers expressed, or not expressed, by the 11 identified clusters of the CD4^+^ TIL population (n = 20, cohort 2). (**C**) Cell frequency of the 11 generated clusters of CD4^+^ TILs (n = 20, cohort 2). (**D**) Cell frequency of the 11 generated CD4^+^ TILs clusters (n = 32, cohorts 2 and 3) according to the MSS/MSI status (pale green and dark blue, respectively). The clusters of the two cohorts were pooled and the cluster numbering from cohort 2 was retained; Mann–Whitney test (* *p* < 0.05). (**E**) Principal component analysis of tumors from cohorts 2 and 3 based on cell frequencies in each generated CD4^+^ TIL cluster, with MSS tumors in pale green triangles and MSI tumors in dark blue circles. Ellipses were drawn to highlight two new groups of CRC tumors: the IC^high^ group (in dark blue) and the IC^low^ group (in pale green).

**Figure 3 cancers-14-04261-f003:**
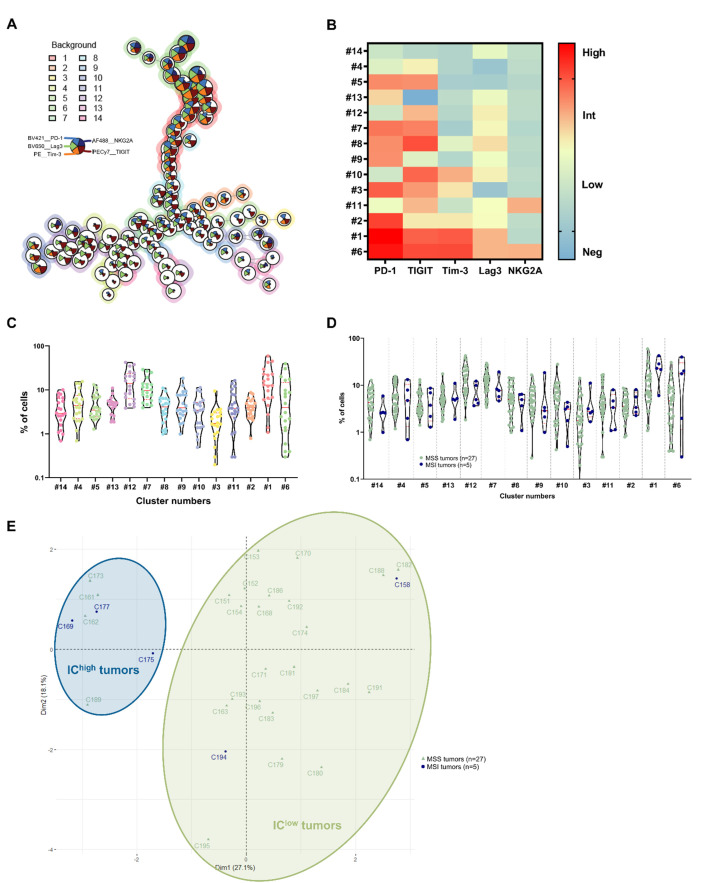
Identification of 14 CD8^+^ TIL clusters based on differential expression of ICs. (**A**) FlowSOM tree of CD8^+^ TILs from cohort 2 tumors (n = 20). The background coloring represents meta-clustering and the legends of the star plot and meta-clustering are shown on the right side. (**B**) Heatmap of the MFI of the PD-1, TIGIT, Tim-3, Lag3 and NKG2A markers expressed, or not expressed, by the 14 clusters identified in the CD8^+^ TIL population generated from cohort 2 tumors (n = 20). (**C**) Cell frequency in each CD4^+^ TIL cluster generated from cohort 2 tumors (n = 20). (**D**) Cell frequency in each CD4^+^ TIL cluster generated from cohort 2 and 3 tumors (n = 32) according to the MSS/MSI status (pale green and dark blue, respectively). The clusters of the two cohorts were merged and the cluster numbering from cohort 2 was retained; Mann–Whitney tests. (**E**) Principal component analysis of tumors from cohorts 2 and 3 based on cell frequencies in each generated CD8^+^ TIL cluster, with MSS tumors are in pale green triangles and MSI tumors in dark blue circles. Ellipses were drawn to highlight two new groups of colorectal tumors: the IC^high^ group (in dark blue) and the IC^low^ group (in pale green).

**Figure 4 cancers-14-04261-f004:**
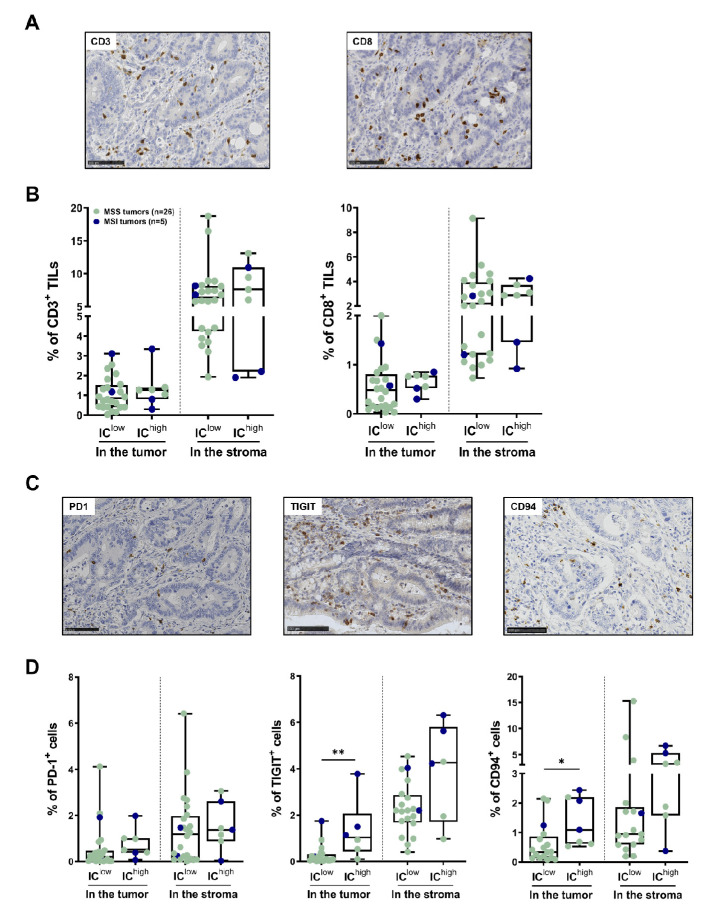
In situ assessment of T-cell density and IC expression profile using immunohistochemistry in the IC^high/low^ subgroups of CRC identified via flow cytometry. (**A**) Representative CD3 (left) and CD8 (right) immunostaining of an MSS CRC with an IC^high^ expression profile. Stained cells appear in brown within tumor glands and in the surrounding stroma; nuclei are counterstained in blue. Bars indicate 100 μm. (**B**) Box plots in the lower panels indicate the percentages of CD3^+^ or CD8^+^ TILs counted inside tumor glands (intraepithelial TILs) or in the peritumoral stroma using QuPath software, according to IC^low^ or IC^high^ subgroups; Mann–Whitney test. MSS tumors are shown in pale green and MSI tumors in dark blue. (**C**) Representative immunostaining of PD-1, TIGIT and CD94 of the same MSS CRC with an IC^high^ expression profile, as in (**A**). Numerous TIGIT^+^ TILs can be seen within the tumor and in the stroma, and only a few PD-1^+^ cells and CD94^+^ TILs. Bars indicate 100 μm. (**D**) Box plots recapitulate the percentages of cells expressing PD-1, TIGIT or CD94, counted inside tumor glands (intraepithelial TILs) or in the peritumoral stroma using QuPath software, according to IC^low^ vs. IC^high^ subgroups; Mann–Whitney test (* *p* < 0.05; ** *p* < 0.01). MSS tumors are shown in pale green and MSI tumors in dark blue.

**Figure 5 cancers-14-04261-f005:**
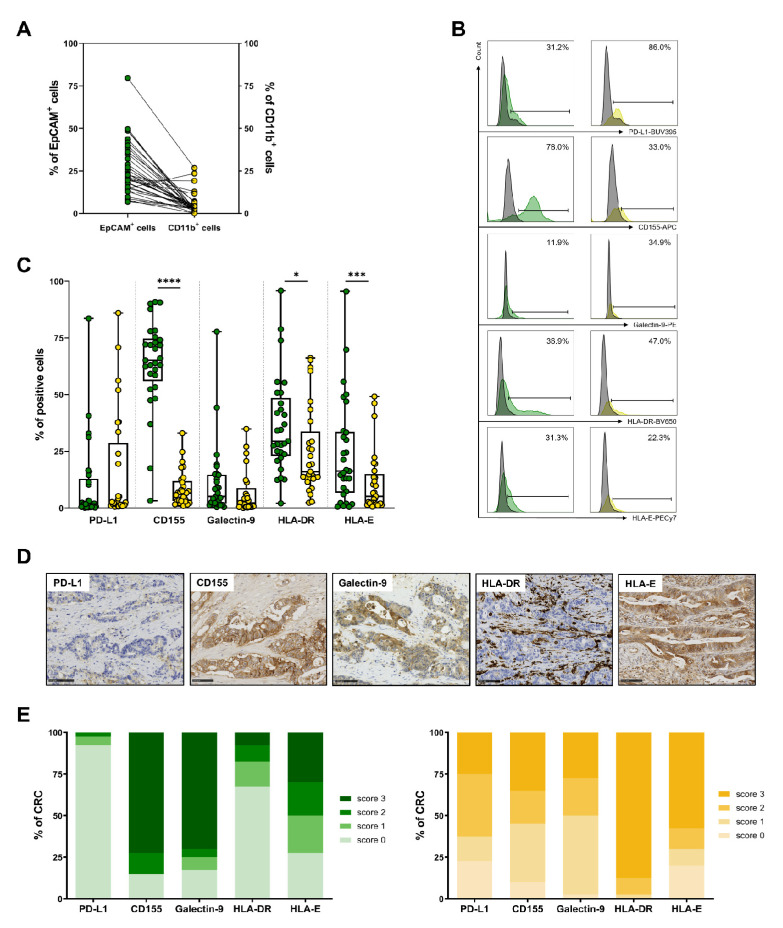
Expression profile of IC ligands by tumor cells and myeloid cells from CRC tumors, determined using flow cytometry and immunohistochemistry. For flow cytometry data (**A**–**C**), EpCAM^+^ tumor cells are shown in green and CD11b^+^ myeloid cells in yellow. (**A**) Paired frequency of EpCAM^+^ and CD11b^+^ cells among viable cells in CRC tumors (n = 29). (**B**) Representative histograms of the expression of PD-L1, CD155, galectin-9, HLA-DR and HLA-E in EpCAM^+^ cells (left) and in CD11b^+^ cells (right). Results are expressed as percentages of positive cells, and isotypic controls are overlaid in grey. (**C**) Frequency of PD-L1^+^, CD155^+^, galectin-9^+^, HLA-DR^+^ and HLA-E^+^ cells among EpCAM^+^ or CD11b^+^ cells (n = 29), determined using flow cytometry; Wilcoxon paired *t*-test (* *p* < 0.05; *** *p* < 0.001; **** *p* < 0.0001). (**D**) Representative immunostaining of IC ligands on an MSS tumor, showing strong expression of CD155, galectin-9 and HLA-E by tumor cells and, conversely, no expression of PD-L1 or HLA-DR. HLA-DR-positive immune cells can be seen surrounding tumor cells. Bars indicate 100 μm. (**E**) Percentages of CRC scoring positive for each IC ligand mentioned on the *x* axis. The semi-quantitative scores used (0 to 3; see Materials and Methods section) appear in green for tumor cells (left) and in yellow for immune cells of the stroma (right).

**Table 1 cancers-14-04261-t001:** Clinicopathological features of CRC patients.

	Cohort 1 (n = 8)	Cohort 2 (n = 20)	Cohort 3 (n = 12)
	7 MSS, 1 MSI	16 MSS, 4 MSI	11 MSS, 1 MSI
Age: mean (range)	61.1 (43–75)	69.8 (40–88)	66.3 (36–85)
Gender			
Men	4	10	9
Women	4	10	3
Tumor location			
Right colon	1	9	3
Left colon	6	9	8
Rectum	1	2	1
pTNM stage (IUCC)			
I	1	3	2
II	4	11	5
III	3	6	5
IV	0	0	0
Histological subtypes			
Adenocarcinoma NOS	6	18	8
Mucinous	2	2	4
CMS classification			
CMS1	3	4	0
CMS2	1	8	3
CMS3	1	5	0
CMS4	1	1	1
Unclassified	0	1	2
ND	2	1	6

pTNM: pathological TNM; NOS: not otherwise specified; ND: not done.

**Table 2 cancers-14-04261-t002:** Semi-quantitative immunohistochemistry of IC ligand expression by tumor cells and immune cells of the stroma in CRC.

Score	Positive Tumor Cells	Positive Immune Cells of the Stroma
0	<5%	Isolated cells or <4 clusters of stained cells/cm^2^
1	5–25%	4–5 clusters of stained cells/cm^2^
2	26–50%	>5 clusters of stained cells/cm^2^
3	>51%	Diffuse staining of numerous positive cells

## Data Availability

The data presented in this study are available in this article.

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
