# Peer review of "Defining the Immune Checkpoint Landscape in Human Colorectal Cancer Highlights the Relevance of the TIGIT/CD155 Axis for Optimizing Immunotherapy"

_cancers, 2022, doi:10.3390/cancers14174261_

Round 1

Reviewer 1 Report

In the present study, Ducoin K et al we evaluated by multiparametric flow cytometry the expression of ICs by TILs in 40 CRC patients and their respective ligands expressed by tumor cells and myeloid cells in 29 patient samples. They found that the majority of CD3+TILs displayed on their cell surface PD1 (77% and 92,4% in MSS and MSI, respectively) and TIGIT (83.3% in MSS vs 90.1% in MSI) whereas lower frequencies of TILs expressed Tim-3 (28.8% in MSS vs 66.9% in MSI), Lag3 (13.3% in MSS vs 21.6% in MSI), and CD94-NKG2A (4.7% in MSS vs 15.3% in MSI). In addition, ex vivo analysis of CRC tumors (n=29, 24 MSS and 5 MSI) showed that tumor cells expressing CD155 (TIGIT-L) were at much higher frequencies than CD155+ myeloid cells (median frequency of 65.2% vs 6.7%). They also found a great heterogeneity among tumor cells expressing HLA-E (frequency range: 0.5% to 95.6% ) which is the ligand for  NKG2A. Moreover, they showed that PD-L1 expression was low both in tumor cells and myeloid cells (2.5% and 1.8%, respectively). Based on their data the authors write in the discussion section: “In light of our results, we believe that the use of anti-TIGIT mAbs should be developed for colon and rectal cancer patients, not in combination with anti-PD-1 or -PD-L1 antibodies, but rather in combination with other IC such as NKG2A according to their expression levels”, and “Our study highlights the TIGIT-CD155 axis as the first to be targeted in therapies using ICi in CRC patients”.

To start with, these conclusions are premature given the limited number (a) of patients analyzed for TILs and myeloid cells and (b) of CRC primary tumors analyzed by immuno histochemistry, but also due to the great heterogeneity of tumor cells expressing the HLA-E  which is the ligand of NGK2A (see above). Moreover, the authors missed a randomized phase 3 trial which showed that frontline pembrolizumab was superior to chemotherapy with respect to progression-free survival in patients with MSI-H–dMMR metastatic colorectal cancer (André, T. et al. Pembrolizumab in Microsatellite-Instability–High Advanced Colorectal Cancer. N. Engl. J. Med. 2020, 383, 2207–2218). In addition, the authors do not provide details for the specificity of the CD94-FITC (BD Biosciences, HP-3D9, RRID:AB396200) used for ICs analyses in the 1st cohort of TILs ( does it recognize NKG2A or the NKG2 family?). I strongly recommend the authors to re-write those parts of the discussion section by adding that their data are only suggestive due to the limited number of patients’ TILs analyzed as well as of the number of tumors analyzed by immunohistochemistry. They should also discuss the aforementioned phase III clinical trial and provide more details for the CD94-FITC.

Furthermore, in their paraffin sections the authors should examine ICs expression levels (a) in relation with the immunoscore classification of CRC as proposed by Galon J and (b) in the context of the tumor areas e.g. tumor center vs invasive margin. It will be interesting to see whether high immunoscores correlate with increased expression of ICs and at which tumor compartments.

Reviewer 2 Report

In the present manuscript, the authors use a comprehensive flow cytometry approach in an attempt to further classify colorectal cancers in terms of their immune checkpoint landscape, and to find new potential targets for therapy. Generally, I find the paper well written, the experimental setup has potential and the results are indeed interesting. I do believe that it is true, that there are subgroups of patients with MSS tumours that have similar immune reactivity as MSI tumours, and that it is an important task to identify these patients.  

In general:

They have an interesting experimental setup but the number of patients is very low, in particular when it comes to subgroup analyses. Unfortunately, this makes these subgroup analyses very uncertain. For example, in total only 5 patients have MSI tumours, and a lot of their findings include this subgroup. The authors need to be very modest in interpreting findings based on small subgroups and avoid terms like slightly increased or statistical trend, when there are no significant differences.

 Even though most of the results are in support their conclusions, there are some over-interpretations, and some over usage of statistical tools to visualize the findings, that need to be addressed.

Comments:

The definition and the subdivision of the patients in to three cohorts is not clear. In general, since the patients are few and the subsets of patients in each cohort very different, I do not see the advantage by presenting data separately for the different cohorts. When possible, and as done in some figures, I would prefer that data was presented for all patients together.

This includes:

Figure 2 and 3, A- C. Results are shown separately for cohort 2 and 3 (there is only one MSI tumour in cohort 3). Why is not cohort 1 included?

The distributions of clusters as shown by UMAP in Figure 2 and 3, E-G, are complicated, difficult to interpret and in my opinion not necessary. The PCA plot clearly defines two different populations of tumours and the distribution of MSI/MSS can be visualised here.

Clusters are divided into MSI/MSS cases. It is interesting to visualise, but hard to draw big conclusions from. Further subdivision of clusters into stage and CMS subgroups are in my opinion not so informative, but okey in the supplements. However, considering Figure S3D-E, why are there only 22 patients (cohort 2 and 3)?

Figure 1E, and the described results from this, I find as an over interpretation. I would suggest the authors to remove this figure.

The in situ evaluation of patient FFPE tumour tissues

The authors evaluate infiltration of intraepithelial T cells and the IC markers in the tumours by immunohistochemistry, but they show only the results from intraepithelial T cells. It would be interesting to see the distribution of IC markers also in CD4 and CD8 positive T cells in tumour stroma.

Results are difficult to interpret due to low sample sizes. The authors further describe that they count three fields within each tumour section, but do not specify the areas, making it hard to determine quality.

The authors find that the tumours express more of CD155, the ligand to TIGIT. Since TIGIT and CD155 are evaluated on the same tumours it would be very interesting to analyse their pairwise distribution. It would also be interesting to analyse the expression of CD155 in groups of IC-low/IC-high.

In Figure 4 B, the lower panel should, according to the figure legend, describe histograms of the percentages of intraepithelial TILs expressing PD1, TIGIT or CD94. Is this also based on immunohistochemistry? In this case, how is this done without double staining? Please elaborate.

How come there are suddenly 7 MSI tumours? Or do the authors rather mean the 7 IC-high tumours?   

Discussion

Previous papers in literature have described and discussed that there are subsets of MSS tumours with similar immune reactivity as MSI, including e.g. POLE mutants. These could be used in support of the findings of this paper.

The main weaknesses of the paper, including a small study cohort, difficulties in interpretation of data from subgroup analyses, and the need for further studies to validate their findings in order to assess a potential clinical relevance, should be thoroughly discussed.

Minor comment:

When describing Figure 2D, the authors accidently wrote in one case: MSI vs MSI instead of MSI vs MSS.

Round 2

Reviewer 1 Report

none

Author Response

This reviewer does not require any modification.

Reviewer 2 Report

Minor comment:

The authors write in the beginning of section 3.4.

“… using immunohistochemistry on paraffin sections from tumors of cohorts 2 and 3 (n=31, 24 MSS, 7 MSI).”

To my understanding, either the numbers are wrong, or they are referring to IC-high, IC-low? Please correct. 
